# Chemical Recycling of PET in the Presence of the Bio-Based Polymers, PLA, PHB and PEF: A Review

**Mohammad Nahid Siddiqui** [1], **Halim Hamid Redhwi** [2], **Abdulrahman A. Al-Arfaj** [1]
**and Dimitris S. Achilias** [3,*]

1. Chemistry Department, King Fahd University of Petroleum and Minerals, Dhahran 31261, Saudi Arabia; mnahid@kfupm.edu.sa (M.N.S.); arfaj@kfupm.edu.sa (A.A.A.-A.)
2. Chemical Engineering Department, King Fahd University of Petroleum and Minerals, Dhahran 31261, Saudi Arabia; hhamid@kfupm.edu.sa
3. Lab of Polymer and Color Chemistry and Technology, Department of Chemistry, Aristotle University of Thessaloniki, 54124 Thessaloniki, Greece
* Correspondence: axilias@chem.auth.gr; Tel.: +30-2310-997822

**Abstract:** The great increase in the production and consumption of plastics has resulted in large amounts of plastic wastes, creating a serious problem in terms of their environmentally friendly disposal. The need for the production of more environmentally friendly polymers gave birth to the production of biodegradable, and more recently, biobased polymers, used in the production of biodegradable or biobased plastics. Although the percentage of currently produced bioplastics is rather small, almost 1% compared to petrochemical-based plastics, inevitably is going to significantly increase in the near future due to strict legislation recently posed by the European Union and other countries' Governments. Thus, recycling strategies that have been developed could be disturbed and the economic balance of this sector could be destabilized. In the present review, the recycling of the polymer mainly used in food plastic packaging, i.e., poly(ethylene terephthalate), PET is examined together with its counterparts from the biobased polymers, i.e., poly(lactic acid), PLA (already replacing PET in several applications), poly(3-hydroxybutyrate), PHB and poly(ethylene furanoate), PEF. Methods for the chemical recycling of these materials together with the chemical products obtained are critically reviewed. Specifically, hydrolysis, alcoholysis and glycolysis. Hydrolysis (i.e., the reaction with water) under different environments (alkaline, acidic, neutral), experimental conditions and catalysts results directly in the production of the corresponding monomers, which however, should be separated in order to be re-used for the re-production of the respective polymer. Reaction conditions need to be optimized with a view to depolymerize only a specific polymer, while the others remain intact. Alcoholysis (i.e., the reaction with some alcohol, methanol or ethanol) results in methyl or ethyl esters or diesters that again could be used for the re-production of the specific polymer or as a source for producing other materials. Glycolysis (reaction with some glycol, such as ethylene, or diethylene glycol) is much studied for PET, whereas less studied for the biopolymers and seems to be a very promising technique. Oligomers having two terminal hydroxyl groups are produced that can be further utilized as starting materials for other value-added products, such as unsaturated polyester resins, methacrylated crosslinked resins, biodegradable polyurethanes, etc. These diols derived from both PET and the bio-based polymers can be used simultaneously without the need for an additional separation step, in the synthesis of final products incorporating biodegradable units in their chemical structure.

**Keywords:** chemical recycling; chemolysis; PET; PLA; PHB; PEF

## 1. Introduction

As it is well-known, plastics are among the most widely produced and used materials in nowadays. According to estimation by Geyer et al. [1], almost 8300 Mt of plastics were produced globally in the period 1950–2015 with nearly half of it (i.e., 3900 Mt) produced

in the last 13 years of this period. This great increase in the production of plastics has resulted in large amounts of plastic wastes, creating a serious problem as far as their environmentally friendly disposal. Landfilling is not an adequate technique since most of the conventional plastics are non-biodegradable and accumulate in the final recipients for many years. Therefore, alternative materials were sought that had similar properties though were more environmentally friendly. This need gave birth to the production of biodegradable polymers used in the production of biodegradable plastics.

In 2019, the global production of bioplastics, was 2.11 million tons. Compared to the total production of plastics, i.e., 368 million tons in the same year, it can be postulated that the current production of bio-based plastics is still rather small, almost 1% [2–4].

At this point, it is worth to clarify the term 'bio' used in bioplastics, biobased, or biodegradable polymers. There is certainly a great confusion in the academic and mainly the industry society. According to *European Bioplastics*, bioplastics are those plastic materials that are biobased, biodegradable or both [5]. The term biobased refers only to the origin of the feedstock, i.e., the source from which the monomer(s) is produced, namely biomass. Thus, the biobased origin does not necessarily determine possible biodegradable properties. These are mainly dictated by the chemical structure of the polymer itself. It is for this reason that petroleum-based polymers, such as polycaprolactone (PCL) are biodegradable. The degradation of a polymer is defined as the chain scission that a polymer undergoes resulting in a decrease of molar mass. The term biodegradable polymer indicates that a material is able to be broken down into carbon dioxide, water, and biomass by the natural action of microorganisms. However, the term by itself does not define how quickly this process will occur, or a specific set of conditions that are required. Therefore, in several polymers it is advisable to use the term compostable instead of biodegradable [5]. 'Compostable', in the context of plastics, is a precisely defined term. It means biodegradation under aerobic conditions into carbon dioxide, water, and biomass within a specific time frame of 6–12 weeks and under specific, controlled conditions, which are specified by the corresponding standard references (ISO 17088, EN 13432/14 995 or ASTM 6400 or 6868) and certification. For example, EN 13432 requires compostable plastics to disintegrate after 12 weeks and completely biodegrade after 6 months, when more than 90% of the organic carbon is converted to $CO_2$. Currently, most of the plastics with the biodegradable tag cannot decompose in conditions found in the natural environment [6–8].

Among the so-called biodegradable polymers, being actually a bio-based and compostable polymer, poly(lactic acid), PLA has received extensive attention in the last 20 years both in academia and in the industry. The biodegradability of PLA depends on the environmental conditions. For example, a PLA bottle may take a few years to decompose in a landfill as opposed to a few months in industrial compost at 60 °C in the presence of digestive microbes [9]. PLA is one of the most promising materials for commercially replacing nondegradable polymers such as poly(ethylene terephthalate) (PET) and polystyrene (PS) mainly in food packaging [10]. Due to similar applications and the lack of an appropriate collection infrastructure, post-consumer PLA waste often contaminates other plastic waste streams (especially PET); thereby, necessitating an additional supplementary effort using advanced sorting technologies in order to achieve a satisfactory reduction [11]. This disturbs the recycling strategies that have been developed for PET and destabilizes the economic balance of this sector [11]. It has been proposed that, even trace contamination of PLA (about 1000 ppm) in conventional PET waste streams renders them unsuitable for mechanical recycling as it causes noticeable hazing and degradation of recycled PET [11,12]. In some cases, this problem has aggravated to an extent that some organizations (viz. NAPCOR) have refused to introduce PLA contaminated PET streams in their existing recycling infrastructure [12].

A comparison of the time evolution of the number of papers published and citing PLA in relation to the corresponding ones on PET is illustrated in Figure 1a. From these results it is clear that 20 years ago, the number of citations to PLA was less than 100 per year while those for PET were nearly 400 for the same year. Both citations present an increase with

time, whereas for PET this increase is nearly linear whereas for PLA almost exponential. In 2020, the number of papers citing PLA was more than 2000 whereas that for PET was nearly 1600. In the same figure the corresponding values for other two biodegradable polyesters, i.e., polyhydroxyalkanoates (PHA) and poly(ethylene furanoate) (PEF) are included. The number of citations to these polymers is rather small, nearly 20–30 per year. It should be noted here that since both abbreviations, PET and PLA are also used to indicate other concepts, the following keywords were used: PET + terephthalate, PLA + lactic, PEF + furanoate and PHA + alkanoate. Given the recent great increase in the interest of the international scientific community on PLA (greater than its counterpart PET) and in the concept of this research, we subsequently searched for the number of papers published on the recycling of these materials. From Figure 1b it is clear that although the interest on PLA recycling has increased in the last 10 years, it is still much lower than that on PET recycling. It seems that the academic interest in PET recycling is very high with increasing trend (almost 300 papers per year) whereas the papers on PLA recycling are still less than 80 and much lower those on the recycling of PHA or PEF. From the combination of the data and the trends presented in Figure 1a,b it is expected that the interest on PLA recycling will definitely increase in the coming years.

The reason for selecting the recycling of PET in this review is based on the fact that currently PET has the largest market volume in bottles for water or beverages and it is also widely used in film applications for food trays and lids with a total world production capacity of over 65 million tons of virgin polymer a year. Of the two monomer reactants for the production of PET, ethylene glycol (EG) can be produced from bio-sources (bio-EG). However, terephthalic acid (TPA) still is a petrochemical derived product. In this context and in order to produce a biomass derived replacement to PET, though being fully bio-based, recently scientists proposed the use of furan-2,5 dicarboxylic acid, FDCA (produced from renewable sugars), with bio-EG to synthesize, poly(ethylene 2,5-furandicarboxylate), or poly(ethylene furanoate) (PEF) [13,14]. In this regard, Eerhart et al. [15] showed that replacing PET with PEF would reduce the non-renewable energy use by 40–50% and the greenhouse gas (GHG) emissions by 45–55% for the cradle-to-grave system. Therefore, large-scale production of bio-sourced PEF will significantly reduce both greenhouse gas emissions and non-renewable energy usage compared to petroleum-sourced PET [15]. However, similar to PET, PEF is not biodegradable or compostable and its end-of-life options must be thus considered to avoid contributing to the accumulation of plastic waste. The papers published on the chemical recycling of PEF are very limited [16].

Bacterial polyhydroxyalkanoates (PHA) is another class of bio-based polyesters used in biomedical applications as bioresorbable materials due to their biocompatibility [17]. They are environmentally friendly which makes them a competitive alternative to synthetic polymers [18]. Poly[(R)-3-hydroxybutyrate)], PHB, is the most common member of polyhydroxyalkanoates, the natural biopolyesters of intrinsic biodegradability and biocompatibility. This material has attracted extensive attention because it has similar thermo-mechanical properties (such as glass transition temperature, melting point, and tensile strength) to conventional commodity polymers, such as polypropylene (PP). Furthermore, PHB has good oxygen permeability and UV resistance and it is water-insoluble and relatively resistant to hydrolytic attack. These properties make it an exception among biodegradable plastics [19].

The two major challenges of humanity today are the fight against the pandemic and the protection of the environment. Concerning the latter, one of the great concerns is the successful management of plastic wastes and particularly plastic packaging. Europe is the leading region in terms of regulations. In 2015, the European Commission adopted the Circular Economy Action Plan which included goals to increase the recycling of the packaging waste and to reduce the landfill by 2030 [20] and a more detailed strategy was laid on 2018 [21]. The reduction of single-use plastic bags had already been targeted [22]. These increasing regulations and bans against single-use plastics are increasing the demand

for biodegradable plastics [23]. However, what will be the impact of bioplastics in the recycling streams of conventional plastics is still under investigation [11].

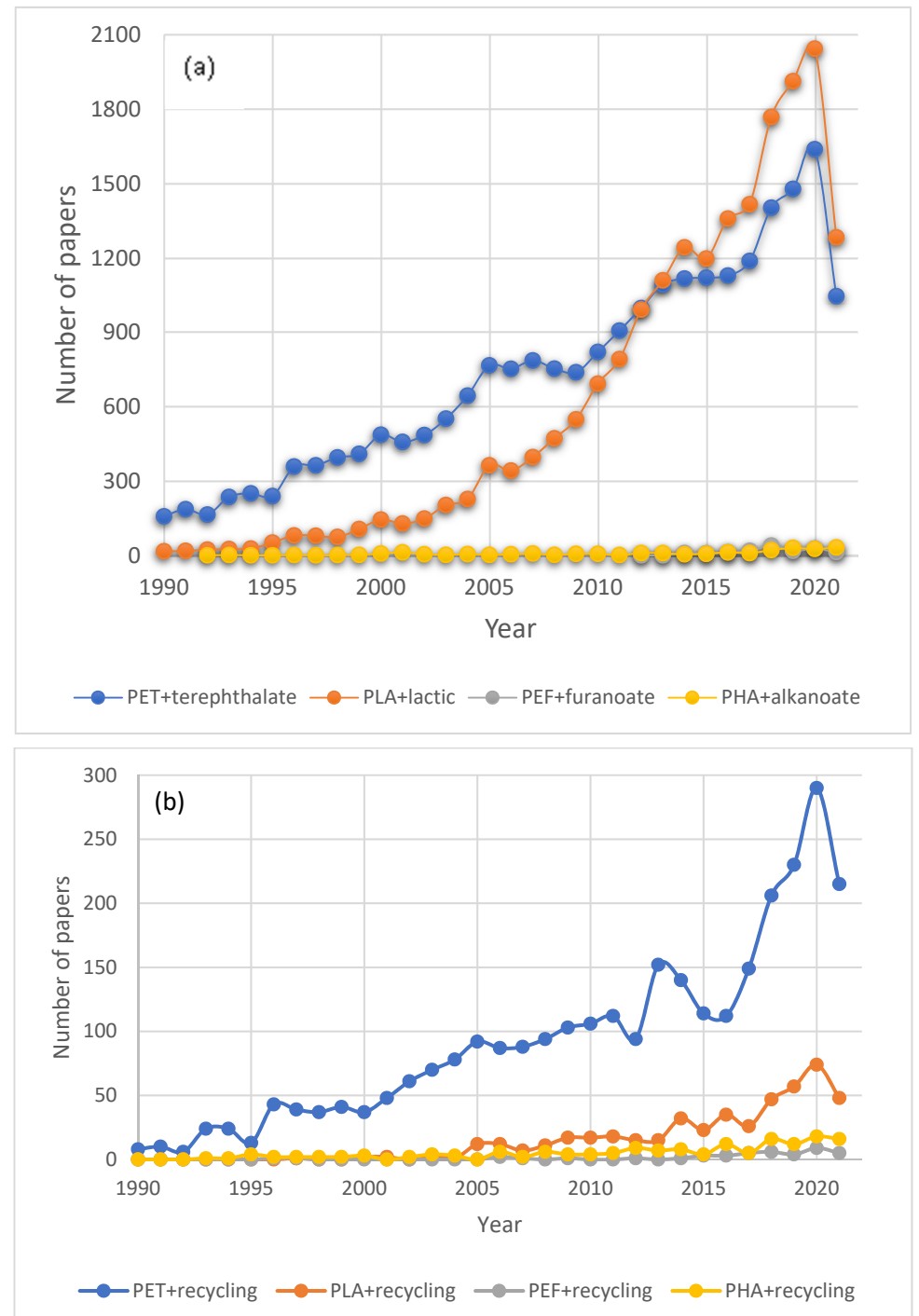

**Figure 1.** Time evolution of the number of published papers citing the polyesters PET, PLA, PEF and PHA (**a**) and the name of the polyester + recycling (**b**) (data from SCOPUS).

In this framework, and keeping in mind that the biobased polyesters PLA, PHA and PEF definitely will replace PET in several packaging applications in the near future, we focus in this work on the impact of their presence on the recycling of PET. Since the risks associated with the presence of these polymers on PET has been already presented in a recent review [11], the aim of this review is to show in detail the chemical recycling characteristics of these plastics. The techniques already reported for the chemical recycling of

the individual plastics based on PET, PLA, PEF and PHB are initially thoroughly reviewed with their advantages and disadvantages. Furthermore, the chemical recycling methods reported for the depolymerization of PET in the presence of PLA, PEF or PHA are also thoroughly examined. The final products (monomers, secondary value-added materials) obtained from each method are compared and analyzed and suggestions are provided on which one would be the best for the recycling of contaminated conventional plastics with bio-plastics.

## 2. Plastics Recycling

### 2.1. General Recycling Routes in Plastic Wastes

In 2020, from the collected post-consumer plastic packaging (17.8 million tons), 42% was recycled, 39.5% was used for energy recovery and 18.5% was landfilled [24]. Most of the waste was recycled mechanically, and only very limited volumes (less than 0.1 million tons) were treated by chemical recycling processes (PlasticEurope, 2019a; PlasticsEurope, 2020) [4,24]. To achieve the circular economy for plastics, zero landfilling is needed. Therefore, the number of polymers recycled should be greatly increased.

At present, the methods applied for commodity plastics' treatment in line with the broader framework for plastic recycling are primary recycling (recycling inside the plant without quality losses), secondary (physical/mechanical/solvent-based) recycling, tertiary (chemical/thermochemical/feedstock) recycling and quaternary (energy recovery) recycling [25] (Figure 2).

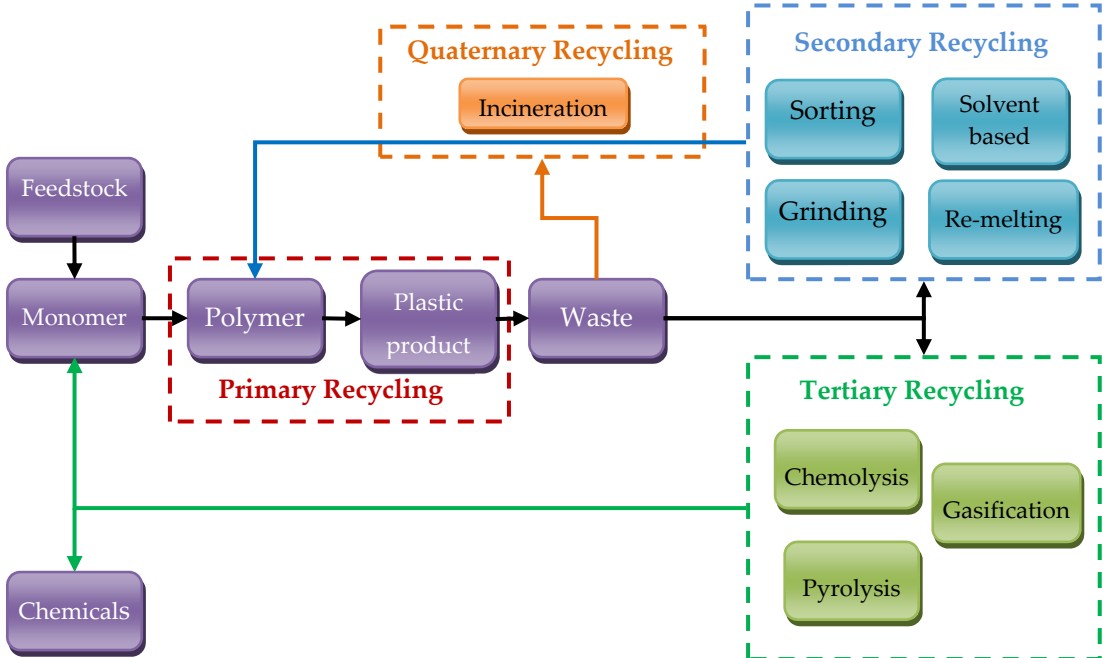

**Figure 2.** Main recycling routes in plastic wastes.

Secondary or mechanical/physical recycling is process in which waste plastics are reprocessed through mechanical means to form polymers, with nearly the same or lower performance in comparison to the initial products. It is a waste to polymer technique. It is mainly applied to homogeneous plastics, so heterogeneous plastic waste needs to be separated and sorted before mechanical recycling takes place. In this category, the solvent-based recycling also falls, where specific solvents are used to dissolve the polymer of interest and further recover it using a re-precipitation technique. This method can be applied to multilayer structures also. Mechanical recycling is employed worldwide and in a large scale for industrial use. Its main drawbacks are the fact that, during every cycle, the product's properties deteriorate [25]. Specifically, their mechanical properties are usually

reduced due to a decrease in the molecular weight owing to chain scissions. This can take place either by increased temperatures and prolonged times during re-melting or the existence of even low amounts of foreign substances such as water. Thus, each polymer can endure only a specific number of reprocessing cycles.

Tertiary or chemical/thermochemical/feedstock recycling is the process in which plastic wastes are converted into lower molecular weight products (such as monomers, secondary valuable chemicals and fuels) through chemical reactions. This is a waste to chemicals technique. Chemical recycling includes various techniques such as hydrolysis, glycolysis, thermolysis, etc. It is considered as an environmentally friendly and economically feasible approach, taking into account that the recovery of monomers or useful chemicals is possible in this way [25].

Finally quaternary recycling refers to the incineration of the plastic wastes and recovery of energy. Actually, this is not a recycling technique but a recovery, since all material is transposed to energy. This is a waste to energy technique.

### 2.2. Recycling of Petroleum-Based Plastics in the Presence of Bio-Based Plastics

The increased number of bio-based plastics mainly in the plastic packaging has emerged the problem of identifying methods for their recycling. Actually, this issue did not exist about a decade ago. However, it has emerged recently and the number of papers on the subject is increasing exponentially. The importance that the existence of bio-based plastics could disturb the current recycling of plastics and could inhibit the potential closure of the life cycle of plastics was examined recently by Alaerts et al. [11]. After extensive research they assessed the risk in three elaborated case studies with great importance in plastic packaging. In this sector, the bio-based polymers increasingly used to replace the main polymer used in food-grade bottles, i.e., PET are PLA, PEF and PHB. Alaerts et al. [11] concluded that future risks are certainly assessed for PLA, whereas for PHB there is no risk currently, but it will be crucial in future application developments. Aldas et al. [26] examined the feasibility of reliably identifying some biopolymers, treated as contaminants, in the recycling process of PET. They noticed a good miscibility between PET-PLA and PET-PHB. However, the problem identified was that PHB degrades in the recycling process of PET due to the high processing temperatures used and the mechanical properties of the recycled PET are decreased due to the presence of PLA or PHB [26]. Indeed, according to data in Table 1, the degradation temperature of PHB is below the processing temperature of PET.

The patent perspective for the recycling of bioplastics has been presented by Niaounakis [27]. In this interesting review several issues have been identified, specifically: Concerning the sorting of bioplastics Near InfraRed (NIR) spectroscopy seems to be the most promising technique for separating PLA from a PET recycling system [28]. However, he noticed that the cost of sorting with an NIR system could be forbidden due to the current low amount of PLA in the plastic waste and no patents dedicated to NIR sorting of biopolymers have been disclosed so far [27]. Concerning the mechanical recycling of the biopolymers using the well-known re-melting technique, again, the point was that its application in the recycling of biobased polyesters such as PHA is not permitted since it is a very thermosensitive polymer (Table 1). Furthermore, reprocessing of PLA is also difficult since it is a hygroscopic polymer and existence of moisture at high temperatures results in its degradation and deterioration of its properties [27]. The presence of small amounts of PLA in the PET waste was found to significantly affect its rheological properties under non-isothermal elongational flow [29]. Thermal stability was not found to be significantly affected [29]. An immiscible blend of PET with PLA is produced [30]. Addition of the compatibilizer ethylene-butyl acrylate-glycidyl methacrylate was found to increase the compatibility of PLA with PET during their recycling [31]. PET in general is incompatible with several biopolymers [32]. The recirculation potential of post-consumer/industrial bio-based plastics through mechanical recycling was studied by Briassoulis et al. [33]. Techno-economic sustainability criteria and indicators were set.

A first review on the recycling methods of bioplastics appeared in 2013 from Soroudi and Jakubowicz [34]. The limitations of the mechanical recycling methods again noticed especially during sorting and thermomechanical degradation. The contamination of the PET waste stream by water bottles made of PLA was pointed and the existence of PLA as a contaminant was highlighted. Even if PLA was sorted using NIR spectroscopy an expensive investment is required which makes the process very costly [34]. The difficulty of separating PLA from PET using differences in density and the sink-float method was also discussed since the density of PLA is similar to PET (Table 1). Different methods for the recycling of bioplastics, including PLA and PHA have been recently reviewed by Lamberti et al. [35].

Due to difficulties associated with the recycling of bioplastics and mainly PLA by mechanical means some researchers have begun to explore chemical recycling methods. Among the first reports on the chemical recycling of PLA was that of Carne Sanchez and Collinson [12]. Alcoholysis of PET and PLA was studied and found to be a good method for the separation of these polymers. At relatively mild conditions PLA was converted to methyl lactate which easily separated from the unreacted PET [36]. Afterwards several papers published on the chemical recycling of PLA mainly that will be reviewed in the next section.

**Table 1.** Physical and thermal properties of some bio-based polymers and PET.

| | Density (kg/L) | Tg (°C) | Tm (°C) | Onset of Degradation (°C) | Processing Temperature |
|---|---|---|---|---|---|
| PET | 1.35–1.39 [11] | 69 [26] | 255 [11] | 410 [36] | 270 [26] |
| PLA | 1.20–1.45 [11] | 45–60 [35] | 155–165 [11] | 320 [37] | 180 [38] |
| PHB | 1.25–1.30 [11,39] | 2–5 [35] | 180 [11] | 220–250 [40,41] | 170 [42] |
| PEF | 1.40–1.55 [11] | 84.3 [16] | 225 [11] | 325 [16] | |

The recycling route of a plastic waste stream including both conventional and bio-based plastics is shown in Figure 3. Briefly the specific plastic products after their production either from fossil-fuel or biomass sources become post-consumer wastes. This mixture could be land filled (which is not preferable) or could be incinerated to produce energy (which is partially beneficial). The more sustainable approach for this waste stream is to be recycled. The first method employed is sorting. In this step bio-polymers could be separated from conventional polymers using different techniques (such as using differences in density, or NIR spectroscopy). However, as it was reported previously, separation of the bio-based polyesters PLA, PHB and PEF from PET is rather difficult since they all have similar densities. Therefore, a mixture of these polymers has to be treated and recycled using the best available technique. These methods of recycling (as in the case of petrochemical based plastics) could be mechanical, solvent-based, chemical, or thermochemical. Using the mechanical recycling method, the mixture could be remelted and reprocessed to form some plastic products. However, as it was reported previously from the properties of the biopolymers compared to PET it seems that rather high processing temperatures are needed for PET which may lead into thermal degradation of PLA or PHB. Solvent-based separation and recycling seems a promising technique as far as environmentally friendly solvents are used. However, it has not been studied in literature for biobased polymers. Thermo-chemical recycling (including pyrolysis and gasification) is very attractive, but mainly useful in the recycling of polyolefins (LDPE, HDPE, PP, etc.) where a mixture of hydrocarbons is obtained that can be used as fuel. However, it is not recommended for the recycling of polyesters, such as PET, since a mixture of organic chemicals are produced which are difficult to be separated. Chemical recycling is an option for bio-based plastics to the same extent as it is for fossil-based plastics. Therefore, chemical recycling seems to be the optimum solution and is discussed in detail in the next section. It should be noted here

that in bio-polymers another method belonging to the chemical recycling methods appears which is enzymatic hydrolysis. However, this was not included in Figure 3 since it cannot be applied to PET.

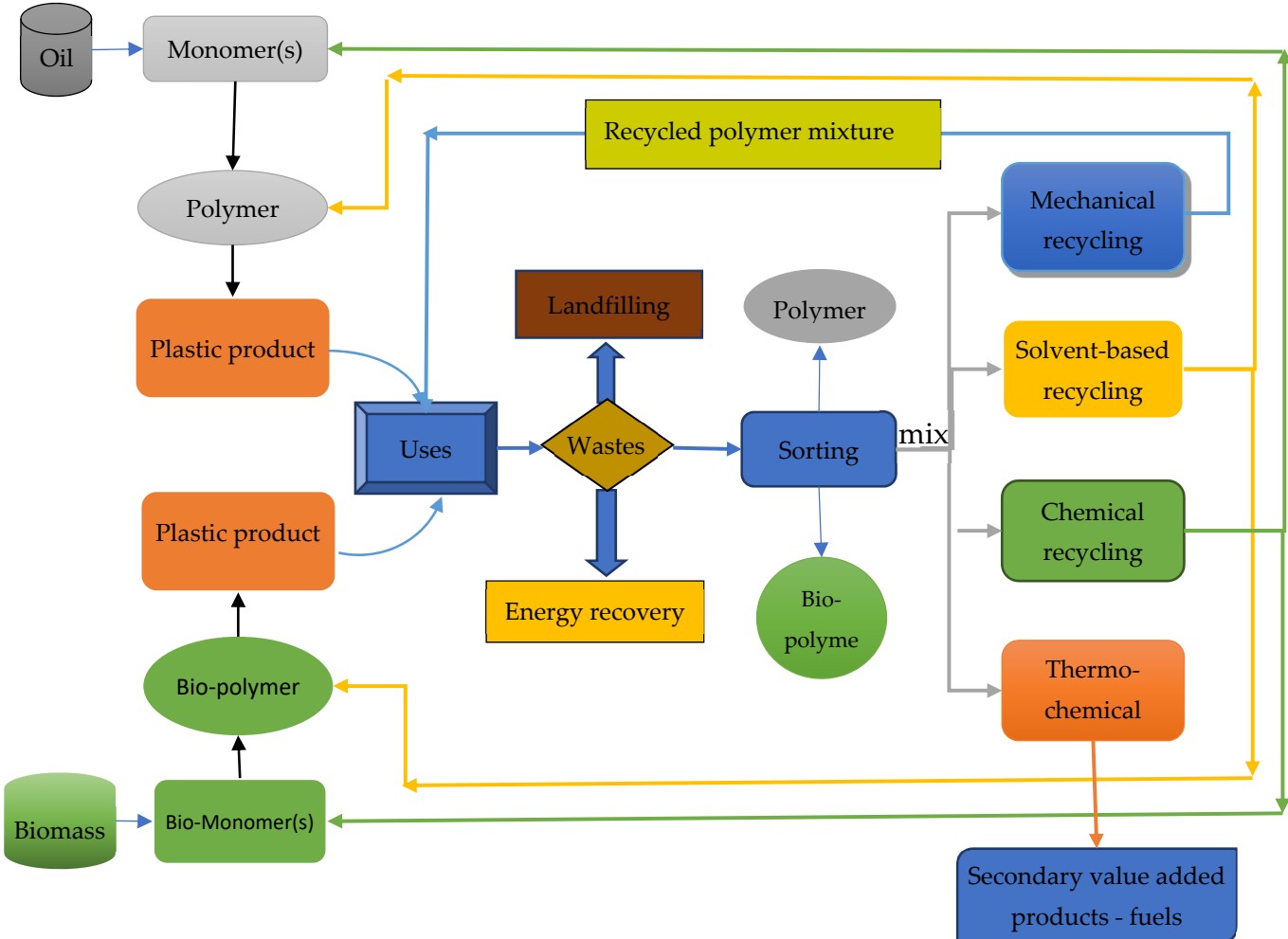

**Figure 3.** Treatment options of a mixed waste stream including petrochemical-based and bio-based plastics.

## 3. Chemical Recycling of Plastics

The EU Waste Framework Directive (2008/98/EC) [43] refers to chemical recycling as any recovery operation by which waste materials are reprocessed into products, materials or substances, which does not include energy recovery or the reprocessing into materials that are to be used as fuels or for backfilling operations. However, the term chemical recycling in the academic literature covers different methods of decomposition of plastic wastes into its constituents (e.g., monomers, oligomers), secondary value-added products (e.g., phenols) or other substances, such as liquid, solid or gaseous hydrocarbons, which could be used as fuels. Degradation of the polymers to its monomers is referred as depolymerization. In research publications two terms, i.e., 'chemical recycling' and 'feedstock recycling' are used sometimes synonymously or differently. 'Feedstock recycling' is defined as a method for breaking the macromolecular chains of the plastic waste to feedstock (i.e., raw material) that could be used for the reproduction of the polymer. 'Chemical recycling' though is also sometimes used to denote the depolymerization of some polymers (i.e., PET) to its monomers. The term chemical recycling is used here as it is more general than that of feedstock recycling.

Chemical recycling describes a process in which plastic wastes are broken down into oligomers or monomers through changes in the chemical structure of the polymers caused

by chemical reactions, usually with the help of heat and catalysts. Chemical recycling allows the plastic material to be recycled again and again, since new virgin polymers can be produced after each depolymerization. It should be stressed here that solvolysis, or solvent-based recycling is usually included in the chemical recycling methods. However, this is not correct since in the solvent-based recycling methods, such as the dissolution and re-precipitation, the chemical structure of the polymer is not affected and specific solvents are only used for the separation of the polymers.

Furthermore, in the research literature two methods are classified in the chemical recycling methods based on the process and the type of the agents used: Chemical and Thermo-chemical recycling. Chemical decomposition (or depolymerization) of polymers is a group of technologies, referred to as 'chemolysis', that use chemical reagents for breaking down polymers into monomers and oligomers [44]. There are different types of chemical depolymerization depending on the type of chemical agent involved: glycolysis, hydrolysis, alcoholysis, etc. [44,45]. The process is capable of obtaining monomers that can be purified by filtering out colorants and additives to produce virgin-grade quality material [46]. Therefore, chemolysis opens opportunities for different industrial applications where pure materials are important, e.g., food contact materials. This technology can only be applied to condensation polymers such as polyesters and polyamides [44] and is mostly suitable for homogeneous plastic waste [47]. In the thermo-chemical recycling, plastic wastes are heated either in the absence of oxygen (pyrolysis or thermal/catalytic cracking) or with limited oxygen (gasification). Pyrolysis is carried out at moderate to high temperatures and results in hydrocarbon compounds resembling to crude oil, which can be treated with conventional refining technologies to produce monomers, other chemicals or fuel-like mixtures. Gasification of plastic waste mainly results in a mixture of hydrocarbons and syngas. Air, steam and plasma can be used as gasifying agents and determine the composition of the syngas, which is used to produce hydrogen and other chemicals. A general classification of the chemical recycling methods is presented in Figure 4.

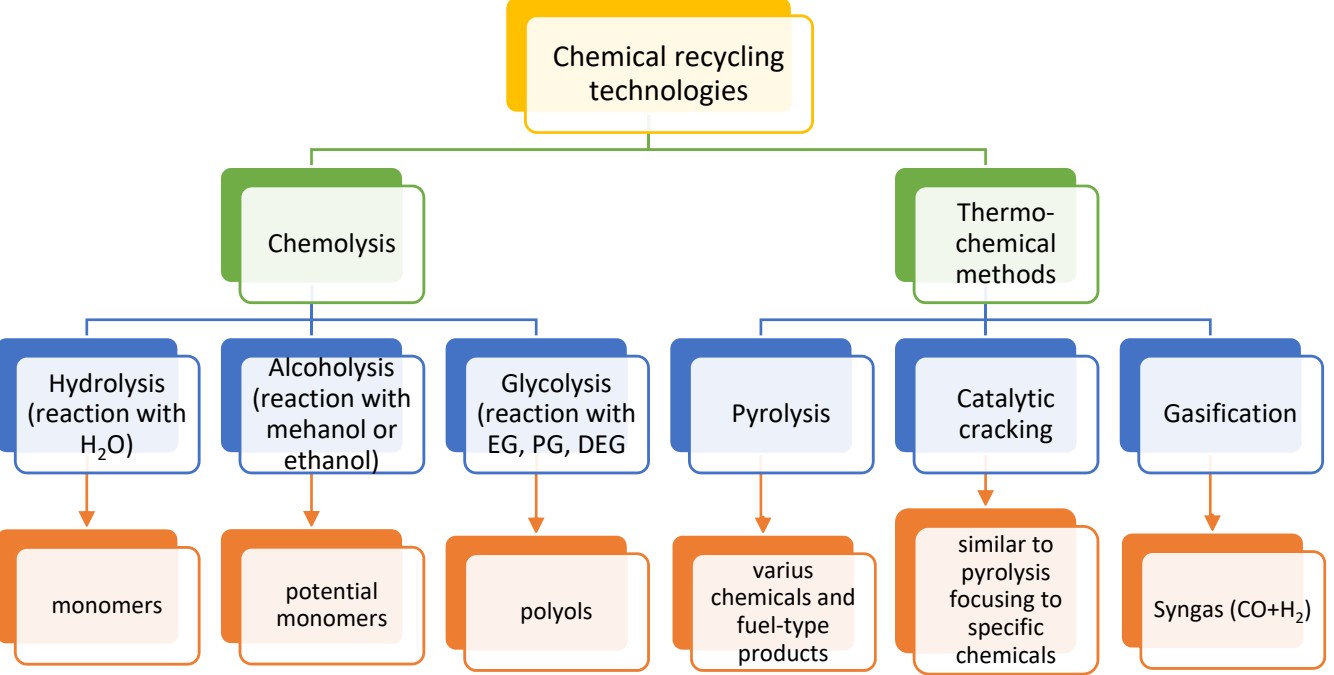

**Figure 4.** Chemical recycling methods.

Thermo-chemical methods (such as pyrolysis, catalytic cracking or hydrogenation) are usually used to recycle polyolefin-based plastics, whereas chemolysis (chemical recycling methods) are preferred in polycondensation polymers. Since the scope of this review is

to present the recycling methods of PET contaminated with bio-plastics such as PLA, PEF or PHA, all these are polyesters and belong to the polycondensation polymers. Therefore, only chemical recycling methods are examined next.

Outputs of chemical recycling are used for the production of plastics or fuels. So, in terms of the European Parliament's (2008) [43] definition of 'recycling', chemical recycling technologies can only in part be classified as recycling. Similarly, the reports by various non-profit organizations highlight that waste-to-fuel schemes in chemical recycling cannot be treated as recycling activities [48,49]. In addition, the role of the chemical recycling in the circular economy is still not clear. The contribution of chemical recycling to the circularity of materials varies. The scholarly literature reviews about the advances of chemical recycling highlight that the products of some applied technologies, such as pyrolysis or gasification, result in the production of fuels or energy, while others, such as chemical depolymerization, could lead to the production of value-added plastic materials [44,46,47].

The chemical recycling of PET has been extensively studied in literature and even several review articles have been published during the last 15 years [50–60]. Review papers on the chemical recycling of PLA have also appear in literature [19,35,61], whereas only a few papers have been published on the chemical recycling of PEF and PHA [16,19]. Different methods for the chemolysis of PET, PLA, PEF and PHB are presented next.

### 3.1. Chemolysis by Hydrolysis

Polyester hydrolysis involves the reaction (-lysis) of the functional ester groups of the polymer by water (hydro-). The process can be carried out in an acidic, alkaline or neutral medium usually at high temperature and pressure, resulting in the formation of the monomers i.e., the diacid and the glycol. Hydrolysis is actually the reverse reaction of the polyester synthesis and as such it is the only method that directly results in the formation of the monomers. Its main disadvantage is the use of high reaction temperatures and pressure as well as long time needed for complete depolymerization.

### 3.1.1. Neutral Hydrolysis

Neutral hydrolysis is carried out using water or steam as reagent. The overall chemical reaction taking place in the neutral hydrolysis of PET, PLA, PEF or PHA is illustrated in Figure 5.

Actually, during hydrolysis of PLA for example, the molecules of water react with the ester bond resulting in two macromolecules, one with a terminal carboxyl and the other with a terminal hydroxyl group. Both react with additional water molecules to finally result in the production of the monomer lactic acid (Figure 6).

The high hydrophobicity of PET hinders neutral hydrolysis, requiring an excess of water or steam at temperatures between 200 and 300 °C and high pressure, 1–4 MPa [55]. The neutral hydrolysis of PET proceeds significantly faster in the molten state than as a solid; therefore, depolymerization at temperatures above 245 °C is highly recommended [55]. High PET hydrolysis rates and complete depolymerization is favored by high temperature, reaction time, high water/PET ratio and use of transesterification catalysts such as alkali metal acetates. Phase transfer catalysts have resulted in lower of depolymerization temperature and time [55]. Recently, a ZSM-5 zeolite was applied as catalyst in neutral depolymerization of PET resulting in a reduction of the energy requirements through carrying out the reaction at lower temperatures [62].

During PET neutral hydrolysis a monoester of glycol with terephthalic acid could be formed as a byproduct. However, it dissolves in water at temperatures more than 100 °C leaving only TPA practically insoluble. Thus, separation of TPA from the reaction mixture is not a problem.

**Figure 5.** Overall chemical reactions taking place during neutral hydrolysis of PET (**a**), PEF (**b**), PLA (**c**) and PHB (**d**) to the corresponding monomer(s) terephthalic acid (TPA) and ethylene glycol (EG), 2,5-furandicarboxylic acid (FDCA) and ethylene glycol, lactic acid (LA) and 3–hydroxybutyric acid (3HBA).

**Figure 6.** Hydrolysis mechanism of PLA to lactic acid.

An obvious advantage of neutral hydrolysis is its environmental friendliness and the avoidance of creation of substantial amounts of inorganic salts (as in the case of alkaline or acid hydrolysis); however, its major drawback which limits its use in industry is that the impurities present in the polyester (i.e., metal catalysts, dyes, pigments, etc.) are not easily separated from the TPA produced during the process. Therefore, TPA has a considerably lower purity than the products of acid or alkaline hydrolysis. Intensive separation and purification of the depolymerized products are necessary in order to obtain TPA of high purity, which makes the entire process time consuming and costly from an economic point of view.

PLA can be hydrolyzed to high percentages of the monomer LA at much lower temperatures compared to PET, i.e., at 160–180 °C [63,64]. The reaction has shown to be autocatalytic, meaning that the carboxyl groups created during hydrolysis further catalyze the reaction. It has been shown that the production of LA from the hydrolysis of PLA is much less energy intensive compared to the production of LA from the fermentation of corn (14 MJ/kg compared to 55 MJ/kg) and also avoids the costly separation needed to separate LA from the fermentation broth [65,66].

Hydrolytic degradation, besides the chemical reaction, also involves the diffusion of water molecules, which begins in amorphous regions of the polyester matrix, and subsequently initiates ester bond cleavage. Hydrolytic degradation continues along the crystalline boundary. A faster rage of degradation was detected when PLA was immersed in 50% ethanol because ethanol molecules diffused within the polymer matrix more rapidly that water molecules causing a swelling of the polymer and facilitating the sorption of water for reaction [67].

The use of water-ethanol mixtures as a medium for PLA hydrolysis has been studied by Auras et al. [67,68]. This study was carried out above the Tg of PLA (40–90 °C) with a 50% ethanol solution. The inclusion of ethanol causes the polymer to swell, facilitating the sorption of water for hydrolysis. Hydrolytic degradation occurred by bulk hydrolysis with the activation energy being lower than for the use of water alone. It was estimated that 41 h would be required to achieve a 95% yield of lactic acid at 90 °C, whereas to form oligomers suitable for repolymerization, 29 h should be sufficient.

Parameters affecting the hydrolytic degradation of PLA besides temperature and time include also crystallinity and average molecular weight of the polymer, pH and solubility of degradation products.

The main differences of PET hydrolysis compared to PLA or PHA is that:

PET is hydrophobic, whereas the presence of hydroxyl groups in PLA makes it more hydrophilic.

PET is completely insoluble in water, whereas PLA is partially soluble.

### 3.1.2. Alkaline Hydrolysis

Alkaline hydrolysis of PET is usually carried out with an aqueous alkaline solution of NaOH, or KOH at a concentration of 4–20 wt%. The reaction products are the disodium or dipotassium terephthalate salt and ethylene glycol, according to the chemical reaction shown in Figure 7 for PET together with PLA, PHB and PEF. The mixture is then heated up to 340 °C to evaporate and recover the EG by distillation, whereas TPA can be obtained by neutralization of the reaction mixture with a strong mineral acid (e.g., H$_2$SO$_4$).

**Figure 7.** *Cont.*

**Figure 7.** Overall chemical reactions taking place during alkaline hydrolysis of PET (**a**), PEF (**b**), PLA (**c**) and PHB (**d**) with NaOH to produce the disodium or sodium salt and neutralization of the corresponding disodium or sodium salt with $H_2SO_4$ to obtain pure terephthalic acid (TPA), 2,5-furandicarboxylic acid (FDCA), lactic acid (LA) or a mixture of 3-hydroxybutyric acid (3HBA) and crotonic acid (CA).

Usually, alkaline hydrolysis of PET takes place at temperatures 210–250 °C, for 3–5 h under a pressure of 1.4–2 MPa. In order to reduce the required degradation temperature, the use of phase transfer catalysts in the form of alkyl ammonium salts has been proposed. Kosmidis et al., [69] performed alkaline hydrolysis mediated by TOMAB (trioctyl methylammonium bromide) at different temperatures; The maximum temperature studied (95 °C) presented the maximum TPA yield as well as the lowest depolymerization time using 5 wt% NaOH with respect to PET.

The main factors influencing the kinetics of the alkaline hydrolysis of PET, include, temperature, depolymerization time, alkali concentration, amount and type of catalyst. Furthermore, alkaline hydrolysis in non-aqueous solutions has been shown to facilitate the degradation. It has been shown that addition of an ether (such as dioxin or THF) accelerated PET degradation.

The main advantage of alkaline hydrolysis is that it can tolerate highly contaminated post-consumer PET wastes and under specific conditions it does not require extreme reaction conditions. However, its main drawback is the production of large amounts of mineral salts, such as sodium sulfate ($Na_2SO_4$) that has to be treated properly.

Chemical recycling of PLA using hydrolysis in an alkaline, NaOH or KOH has been reported in literature [70–74]. Elevated temperatures and subcritical conditions are usually required to achieve high reaction rates. Yagihashi et al. reported the effective conversion of PLLA to l-lactic acid under alkali NaOH, hydrothermal conditions. The conversion was almost complete within 20 and 60 min at 453 and 343 K, respectively. At 343 K, NaOH concentrations above 0.6 M hardly affected the reaction rates. According to the authors, the degradation reaction took place on the surface of the polymer and the dissolution of the products on the pellet surface controlled the degradation reaction [70].

In another study Tsuji and Ikada performed the hydrolysis of PLLA films in 0.01 N NaOH at 37 °C [71]. The authors evaluated the change in molecular weight distribution

and surface morphology of PLLA films during hydrolysis and concluded that hydrolysis is principally done via the surface erosion mechanism mainly in the amorphous region. The rate of weight loss per unit surface area showed a linear decrease with the increasing crystallinity of the films. Microwave irradiation was employed to a phase transfer catalyzed alkaline hydrolysis of PLA to facilitate the depolymerization process [72]. The required degradation time and temperature were much less compared to PET depolymerization under the same experimental conditions.

The use of basic conditions leads to a random chain scission via back biting reactions to generate LA, which is subsequently hydrolyzed [73,74]. Accordingly, the nature of the hydroxyl terminal can dictate the kinetics of this process, with capping of the alcohol preventing this mechanism from operating.

The hydrolysis of PHB does not need high temperature and high pressure, but a large amount of inorganic acid or alkali was used as catalyst. They need to be neutralized, washed and other operations, which leaded to process cumbersome, equipment corrosion, environmental pollution and other issues [75]. Moreover, these catalysts cannot be reused for the next time [76]. Hydrolysis of PHB was investigated in acid and alkaline solutions by monitoring the formation of two monomer products, 3-hydroxybutyric acid (3HBA) and crotonic acid (CA) (Figure 7) [75,77]. Different behavior was observed regarding the tolerance of PHB to degradation under alkaline or acidic conditions. In alkaline solution (0.1–0.4 N NaOH), the hydroxyl ion plays the role of a reagent rather than a catalyst. A high concentration of the hydroxyl ions promoted the hydrolysis rate of the polyester. At 70 °C, 70% of PHB was decomposed at 4 N NaOH. Both CA and 3HBA are produced. However, in acid hydrolysis, ($H_2SO_4$), the protons play the role the catalyst for both hydrolysis and esterification (repolymerization). Therefore, newly formed carboxylic acid and alcohol from an ester bond hydrolysis can be re-esterified by protons. Re-esterification is favored through the stereo-orientation and proton catalysis.

The hydrolysis of the PHB in concentrated acidic solutions (80–98%) generated mainly CA (98%) and a small amount of 3HBA (2%). CA resulted from the dehydration of 3HBA in concentrated acid solutions. CA was shown to be a stable hydrolytic product in concentrated acid solution.

The impact of molecular weight and morphology upon the hydrolytic degradation of PHB was explored by Bonartsev et al. [78]. The authors investigated the hydrolytic degradation in phosphate buffer (pH = 7.4) at 37 °C and 70 °C. The low molecular weight polymers (Mw = 50 and 170 KDa) showed higher degradation rate.

PEF powders of various molecular weights (6, 10 and 40 kDa) were synthetized and their susceptibility to enzymatic hydrolysis was investigated for the first time. According to LC/TOF-MS analysis, cutinase 1 from *Thermobifida cellulosilytica* liberated both 2,5-furandicarboxylic acid and oligomers of up to degree of polymerization 4. The enzyme preferentially hydrolyzed PEF with higher molecular weights but was active on all tested substrates. Mild enzymatic hydrolysis of PEF has a potential both for surface functionalization and monomers recycling [79].

It has been shown that PEF-based macrocycles can be recovered when taking advantage of the cyclodepolymerization of PEF at highly diluted conditions. These macrocycles can subsequently be repolymerized into virgin-like materials [80]. While this process can be carried out on a laboratory scale, its implementation on large scales still needs further development, as much more diluted conditions are required to obtain high yields.

### 3.1.3. Acid Hydrolysis

Acid hydrolysis involves the reaction of the polyester with a mineral acid, usually sulfuric acid, or nitric or phosphoric acid to produce the monomers. The overall chemical reaction taking place during acid hydrolysis of PET, PLA, PEF or PHA is similar to that occurring during neutral hydrolysis illustrated in Figure 5.

The reaction proceeds by a dissolution of the polymer in a concentrated acid followed by heating. In order to avoid high temperatures usually concentrated acids are used (e.g.,

$H_2SO_4$ with concentration > 14.5 mol/L). The production of pure TPA by acid hydrolysis of PET in a 90% $H_2SO_4$ solution at 85–90 °C has been described. It has been established that acid-catalyzed depolymerization follows the shrinking-core model, resulting in lower TPA yields for similar reaction times when larger particles are reacted [55].

Parameters affecting acid hydrolysis include depolymerization temperature and time together with acid concentration and PET particle size.

The main advantage of the process is that pure TPA and EG is produced. However, one of the drawbacks of this kind of process is carbonization of the products (especially EG) that can be induced by the strong dehydrating effect of sulfuric acid, normally decreasing the reaction yield. By using nitric acid instead of sulfuric acid, decarbonization can be avoided. However, due to the oxidizing nature of nitric acid, sodium sulfate ($Na_2SO_4$) is incorporated to minimize the oxidative effect. In addition, there is a need to recycle large amounts of concentrated acids.

Under acidic conditions, the PLA hydrolysis has been shown to be dominated by a chain-end scission, whereby the terminal hydroxyl group is activated by protonation and is hydrolyzed directly to lactic acid [81–83]. In keeping with this mechanism, the rate of degradation was observed to be independent of chain length and this preference was due to the hydrophobicity of the polymer chain compared to the increased hydrophilicity of the chain end [61].

### 3.2. Chemolysis by Alcoholysis

The alcoholysis of polyester consists in breaking the backbone ester linkage through an attack of alcohol (e.g., methanol or ethanol) on the carbonyl group, known as a transesterification reaction, leading to the formation of a dialkyl terephthalate and ethylene glycol as main products for PET. The overall methanolysis of PET to dimethyl terephthalate (DMT) and EG is illustrated in Figure 8. Similarly, methanolysis of PEF results in the production of dimethyl furanoate (DMF) and EG. Both DMT and DMF after hydrolysis can be converted to the corresponding monomers TPA and FDCA (Figure 8). For PLA the reaction with either methanol or ethanol results in the production of methyl lactate (MeLa) or ethyl lactate (EtLa) respectively (Figure 8). Finally, methanolysis of PHB results in the production of methyl 3-hydroxybutyrate (M3HB) shown also in Figure 8. Several acetate catalysts, i.e., zinc, magnesium, cobalt, etc. enhance the reaction.

The uncatalyzed reaction for PET methanolysis requires severe conditions, e.g., pressures of 2–4 MPa and temperature of 180–280 °C and sometimes use of supercritical methanol ($T_c$ = 512.3 K, $P_c$ = 8.09 MPa) in order to achieve high yields of DMT [57].

Recently, new green solvents, such as ionic liquids (ILs) and deep eutectic solvents (DES) have been proposed as catalysts. The depolymerization of PET catalyzed by ILs and DESs holds a few advantages over the use of conventional catalysts such as heavy metal salts: the reaction can proceed at lower pressure, the catalyst can be recovered and is reusable, and the product can be easily purified [50].

Advantages of methanolysis of PET include the production of DMT with quality identical to virgin DMT, EG and methanol can be easily recovered and recycled. Thus, the products can be used for the re-production of the polymer. The disadvantages of the process include the high cost associated with the separation of the reaction products, the highly energy demand when supercritical methanol is used, and mainly the use of TPA nowadays in the production of PET instead of DMT. This adds an extra cost to the process from the hydrolysis of DMT to TPA.

Alcoholysis of PLA has been most commonly performed with methanol or ethanol to give methyl lactate (MeLa) or ethyl lactate (EtLa) (Figure 8). Both can be used as green solvents, with good biodegradability and low toxicity, or as a potential platform to other chemicals [19,61]. For example, alkyl lactates can be transformed into lactide, thus "closing the loop" on the PLA life-cycle. Catalysts used in the alcoholysis of PLA include acids ($H_2SO_4$), solid acids (Montmorillonite K10, CaO), ferric chloride ($FeCl_3$) $ZnCl_2$, $Zn(OAc)_2$, $Zn(Octanoate)_2$, $AlCl_3$, NaOAc, NaOH and NaOMe [19,61]. Temperatures used are usually

in the boiling point of the alcohols but frequently much higher temperatures were used, ranging from 130 to 210 °C, whereas reaction time ranges from 1 to 4 h. A range of solvents have been disclosed including ionic liquids, alcohols, toluene, alkyl lactates and chloroform [19,61]. In the patent literature, there are also several examples of alcoholysis reactions of PLA [27].

Main process parameters are the degradation temperature and time, the type and amount of catalyst and solvent used, as well as the type of the alcohol used

Compared with PLA, depolymerization of polyhydroxyalkanoates is significantly less studied in literature. Actually, recycling of poly(3-hydroxybutyrate), PHB, was mainly studied by Liu et al. [84–87], the rest of the researchers target the degradation of PHB to obtain value-added building blocks. These authors investigated different parameters to optimize the production of methyl 3-hydroxybutyrate (M3HB). Different ionic liquids were used as catalysts instead of strong acids or alkalis used by other researchers that need neutralization and washing operations, resulting in equipment corrosion and environmental pollution [88,89]. Optimum conditions included a temperature of 140 °C, depolymerization 3 h, and amount of methanol to PHB equal to 5:1.

PEF like other polyesters, can be depolymerized by solvolysis. For example, De Jong et al. [90] investigated the methanolysis of PEF in the presence of sodium methoxide/methanol solution at 90 °C, but only moderate yields were obtained (~60%).

### 3.3. Chemolysis by Glycolysis

Glycolysis is the degradation of a polyester by some glycol (e.g., ethylene glycol, diethylene glycol, propylene glycol, etc.) in the presence of a trans-esterification catalyst, mainly metal acetates, where ester linkages are broken resulting in hydroxyl terminal groups. Indicative reactions of PET, PLA, PEF or PHB with either EG or DEG appear in Figure 9.

**Figure 8.** *Cont.*

**Figure 8.** Overall chemical reactions taking place during alcoholysis of: (**a**) PET and (**b**) PEF with methanol to produce dimethyl terephthalate (DMT) and dimethyl furanoate (DMF) and hydrolysis of the products to produce the corresponding monomers, (**c**) PLA with methanol or ethanol to produce methyl lactate (MeLa) or ethyl lactate (EtLa) and (**d**) PHB with methanol for the production of methyl 3-hydroxybutyrate (M3HB).

Glycolysis of PET is one of the oldest and simplest methods used worldwide for the recycling of PET in industrial-level due to its mild reaction conditions and less volatile reagents (e.g., EG, DEG) compared to alcoholysis [50]. Commercial units have already been set up by DuPont, Dow Chemicals, Shell Polyester, etc. [91,92]. Glycolysis involves the transesterification of PET with an excess amount of glycol, at temperatures ranging from

170 to 300 °C to produce oligomers, such as bis-hydroxyethyl terephthalate (BHET) with two terminal hydroxyl groups [52]. The main process parameters are the type and concentration of the glycol used as reagent (e.g., ethylene glycol (EG), diethylene glycol (DEG), propylene glycol (PG), dipropylene glycol (DPG) and 1,4-butanediol (BD)), glycolysis time, temperature and pressure [93]. However, the main parameter affecting the depolymerization rate is the type of the catalyst used and several papers have been published on its selection [94]. Typical catalysts are metal acetates, such as Zn, Mn, Co, Pb acetates, metal chlorides, carbonates, bicarbonates and sulphates, Nitrogen-based organocatalysts, polyoxometalates, a wide range of asic, acidic and neutral ionic liquids, deep eytectic solvents, metal oxides ($ZnO$, $Co_3O_4$, $Mn_3O_4$), magnetic nanoparticles ($\gamma$-$Fe_2O_3$), Mg-Al hydrotalcites, etc. [94].

Figure 9. *Cont.*

**Figure 9.** Overall chemical reactions taking place during glycolysis of PET (**a**), PEF (**b**), PLA (**c**) and PHB (**d**) with ethylene glycol or diethylene glycol to produce different oligomeric diols.

The advantages of glycolysis include: operation at relatively low temperature (<190 °C) under atmospheric pressure, and in a continuous process [95]; the reagents and products are non-toxic with low volatility; no acid, alkali or mineral salts are generated during the process; the products can be easily separated and purified by hot water extraction, cooling crystallization and adsorption; the oligomers formed can be used to produce rPET or as a raw material for the production of several secondary value-added products, such unsaturated polyester resins, polyurethane foams, etc. Among the disadvantages of the process are the relatively long reaction time and low BHET yield, which poses challenges to its large-scale commercial application [96].

Glycolysis of PLA was not studied so far in literature as a method for the recycling of PLA. There are only few reports on the evaluation of products obtained from the reaction of PLA with ethylene glycol. After reaction for more than 30 min at its melting temperature of 170–195 °C, Hydroxyl-terminated oligomers were obtained which used either as a macromolecular cross-linker for epoxidized natural rubber (ENR) or curable precursors for production of thermosetting (co)polyesters [97–99]. In a recent work, the microwave-assisted glycolysis of PLA with EG, propane-1,3-diol (PDO), butane-1,4-diol (BDO) and tetrabutyl orthotitanate (TBT) as a catalyst was investigated. Microwave irradiation is employed as a heating source for shortening the reaction time and improving the reaction efficiency. The products are then utilized as starting materials for other value-added products, especially degradable lactide-based polyurethanes (PUs) [100].

The depolymerization of PEF film waste was carried out by glycolysis using bio-EG and a thermally stable acid-bas organic catalyst system [16]. The procedure followed was similar to that used for the glycolysis of PET by Jehano et al. [101], heating at 180 °C for 2 h and the product of the reaction bis(2-hydroxyethyl) furanoate (BHEF), was similar to the bis(2-hydroxyethyl) terephthalate (BHET) obtained from the depolymerization of PET. BHEF was repolymerized into PEF by melt polycondensation followed by SSP.

## 4. Discussion

The need to replace petrochemical-based polymers to more environmentally friendly material emerged the production of biodegradable and recently bio-based polymers. Plastics made of bio-based polymers are gaining the global market mainly as plastic packaging. These materials eventually become post-consumer wastes that are usually mixed with petrochemical-based plastics. Although currently the amount of biobased plastics is less than 2% of the total plastic waste, their existence in the plastic waste stream may affect the already applied recycling technologies. In this review paper the effect of the existence of such biobased polymers on the chemical recycling of commodity petrochemical-based polymers was studied. Particularly, the presence of the polyesters PLA, PHB and PEF in the PET recycling process was examined, since these polymers are used in applications similar to those of PET in plastic packaging and they are going to replace PET in the near future. The methods used for sorting of these materials are rather limited since they have physical properties (e.g., density) similar to PET. Moreover, their presence in mechanical recycling processes (i.e., re-melting) is not recommended since they decompose at temperatures needed for the re-processing of PET. Thus, chemical recycling seems to be the best solution. Chemical recycling of bio-based plastics contributes to a circular bioeconomy by keeping renewable content in the loop even longer, requiring not only fewer fossil resources, but also less renewable feedstock to produce new, high-quality bio-based plastics.

From the literature review it was clear that chemical depolymerization of PET and PLA have been extensively studied, though the number of articles on the chemical recycling of PHB and PEF is limited. It should be pointed that some methods, such as solvent-based recycling, were not included since they do not alter the chemical structure of the macromolecular chains and do not belong to typical chemolysis. Moreover, enzymatic hydrolysis was also not examined since it is not a recycling method but rather a depolymerization method. Finally, thermochemical recycling methods, such as pyrolysis or gasification, were also not investigated since valuable products, such as mixtures of hydrocarbons, etc., are only produced when polyolefins are used as feedstock or in general polymers obtained from chain polymerization. In contrast, in polycondensation polymers, as the polyesters studied here, the pure chemical depolymerization methods are more adequate. Specifically, hydrolysis, alcoholysis and glycolysis were analyzed.

Hydrolysis carried out under different conditions (acidic, alkaline neutral) and catalysts is applied for the re-production of the monomers, i.e., terephthalic acid, ethylene glycol, lactic acid, or 2,5-furandicarboxylic acid mostly at pre-consumer level that potentially could be used for the re-production of the specific polymer. However, adequate separation processes should be developed. Moreover, the process conditions could be optimized with a view to depolymerize the biobased polymer(s) to produce its monomer(s) while PET could remain intact. Thus, the development of an efficient recycling process for biodegradable polymers requires fast reaction rates and selective cleavage of the chemical bonds during the decomposition process under mild conditions.

Alcoholysis with methanol or ethanol results in products such as dimethyl terephthalate, dimethyl furanoate or diethyl terephthalate, diethyl furanoate for PET or PEF respectively and methyl lactate or ethyl lactate for PLA. The latter could be used as secondary value-added materials whereas the first as potential monomers for the reproduction of the polymers.

Glycolysis, meaning reaction with some glycol (ethylene, diethylene propyl, etc.) although extensively studied for PET is not much investigated so far for the chemical

recycling of biobased polymers such as PLA, PEF or PHB. During glycolysis oligomers with two terminal hydroxyl groups are produced. The products can be further utilized as starting materials for other value-added products, such as unsaturated polyester resins, methacrylated crosslinked resins, polyurethanes, etc. Hydroxyl-terminated oligomers derived from both PET and the bio-based polymers can be used simultaneously without the need for an additional separation step in the synthesis of final products incorporating biodegradable units in their chemical structure. Therefore, glycolysis seems to be the most promising chemical recycling method for treating mixtures of biobased with petrochemical based polymers and particularly polyesters of PLA, PEF and PHB with PET.

Overall, although hydrolysis seems to be the most direct method for the recovery of the monomers it is not recommended for the recycling of PET "contaminated" with bio-based plastics, since different experimental conditions are needed and several additional separation steps are needed. For similar reasons also alcoholysis with methanol, or ethanol is not recommended due to several drawbacks coming from different products obtained. From this review, glycolysis is suggested as the most appropriate method for the specific recycling, since in the products obtained, the key factor is their character as diols (i.e., materials with two terminal hydroxyl groups), whereas their main backbone chemical structure is not so important. The mixture of diols produced could be used for the synthesis of several value-added products such as, biodegradable polyurethanes, thermosetting (co)polyesters, alkyd resins, etc. Additional research is needed in this direction to highlight the advantages of the method and potential problems that will arise.

As a future outlook, the study of glycolysis of PET in the presence of PLA (mainly), but also PEF and PHB, to investigate the effect of the reactants and the reaction conditions on the depolymerization and the quality of the products is proposed. The potential applications of the glycosylates could be studied and the formation of novel value-added materials could be explored. Finally, further research is needed on the development of specialized catalysts for the targeted production of specific products during chemical recycling of PET in the presence of PLA. PHB or PHF.

**Author Contributions:** Conceptualization, methodology, validation, formal analysis, investigation, data curation, writing—original draft preparation, D.S.A.; writing—review and editing, M.N.S., H.H.R. and A.A.A.-A.; project administration, M.N.S., H.H.R. and A.A.A.-A.; funding acquisition, M.N.S. and H.H.R. All authors have read and agreed to the published version of the manuscript.

**Funding:** Authors would like to thank Deanship of Scientific Research, King Fahd University of Petroleum and Minerals (KFUPM) for funding this work through project number DF191041.

**Data Availability Statement:** Not applicable.

**Acknowledgments:** Authors would like to acknowledge the support provided by the Chemistry department, King Fahd University of Petroleum and Minerals (KFUPM) and Department of Chemistry, Aristotle University of Thessaloniki, in carrying out this work.

**Conflicts of Interest:** The authors declare no conflict of interest.

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
