# Peer review of "Chemical Recycling of PET in the Presence of the Bio-Based Polymers, PLA, PHB and PEF: A Review"

_sustainability, doi:10.3390/su131910528_

Round 1
Reviewer 1 Report
The manuscript represents a well written review devoted to the urgent topic in the field of polymer materials recycling. The current state of art in the field with the appropriate citation is comprehensively discussed in the text. The manuscript is logically structured and well illustrated. The work is really nice and merits to be published in Sustainability.
As a minor issues I would suggest the following corrections:
Figures 5-9. Please insert letters a, b, c and d to designate the reactions with certain polymer. In figure legend, please, indicate them as well. For example: “Figure 5. Overall chemical reactions taking place during neutral hydrolysis of PET (a), PEF (b), PLA (c) and PHB (d)…
Also, please replace in the figure captions “PHA” with “PHB” illustrated in fact in Figures 5-9.
Short paragraph on Future Outlook is highly desirable to complete the nice discussion on the chemical recycling of industrial polymers.
Author Response
Dear Reviewer,
Thank you very much for your kind comments.
Figures 5-9 have been changed according to the suggestion. Thank you.
The abbreviation PHB has ben changed in several parts of the manuscript.
A short paragraph with Future Outlook has been added in the last paragraph of the Discussions section.
Reviewer 2 Report
This review article describes about chemical recycling of PET in the presence of the bio-based polymers comprehensively, therefore, this is suitable for publication. But some mistakes and errors should be corrected, and the manuscript also should be re-checked. This manuscript is very long, therefore, some mistakes and errors might not have been checked. I will show some examples, but I think another mistakes and errors seem to be found.
Line 71: Compostable’ seems to be ‘Compostable’ or Compostable.
Figure 1: In the caption, (a) and (b) are written, but these seems to be missing in the figure.
Figure 3: “Waste” in the figure seems to be wrong (this is written as vertical writing).
Line 521: P(3HB) seems to be PHA. If P(3HB) means another plastic, the non- abbreviated name should be indicated.
Line 523: Figure 6 seems to be Figure 7.
Line 619: As shown in Figure 8, (Me-La) and (Et-La) seem to be (MeLa) and (EtLa), respectively.
Line 633: “poly(3-hydroxybutyrate), PHB,” seems to be “PHB”, and I think “poly(3-hydroxybutyrate)” seems to be written when "PHB" appeared first.
Line 592 and 636: M3HB was written as methyl hydroxyl butyrate at line 592, but M3HB was written as methyl 3-hydroxybutyrate at line 636. These are same, but I think terms should be unified. Also, in Figure 8, “3” of M3HB was written as subscript.
Author Response
Dear reviewer,
Thank you very much for your kind comments.
We have re-checked the whole manuscript for minor mistakes, such as those that you mention in your report, and we have revised the text properly, as it can be seen in the re-submitted version.
(a) and (b) have been included in Figure 1
Figure 3 has been modified properly
PHB has been changed throughout the manuscript.
Thank you again for the carefull reading.